# Molecular Dynamics Study on the Effect of SiO$_2$/Al$_2$O$_3$ Mass Ratio on the Structural Properties and Viscosity of Molten Fused Red Mud

**Bo Xu, Yaran Cao, Zhengzheng Wang, Peipei Du and Yue Long ***

School of Metallurgy and Energy, North China University of Science and Technology, Tangshan 063009, China;
boxuchina@163.com (B.X.); acaoyaran@163.com (Y.C.); zheng1225764712@163.com (Z.W.);
1710508@stu.neu.edu.cn (P.D.)
* Correspondence: longyue@ncst.edu.cn

**Abstract:** In this study, molecular dynamics simulation was used to study the effect of SiO$_2$/Al$_2$O$_3$ mass ratio on the structural properties and viscosity of molten fused red mud. The stability of various T–O bonds in the melt was elucidated by analyzing the bond angle and coordination number; the degree of polymerization, and the stability of the melt were explored by analyzing the number of T–O–T bridging oxygen (BO) and the distribution of $Q_{Si}^n$ and $Q_{Al}^n$ of [SiO$_4$]$^{4-}$ as well as that of $Q_{Si}^n$ and $Q_{Al}^n$ of [AlO$_4$]$^{5-}$; the self-diffusion coefficient of each atom was determined by mean square displacement (MSD) analysis; and the trend of the melt viscosity was analyzed according to the relationship between diffusion and viscosity. The results show that as the ratio of SiO$_2$/Al$_2$O$_3$ increases, the viscosity of molten fused red mud first increases, then decreases, and finally increases. This is because Ti$^4$ and Fe$^{3+}$ combine with O$^{2-}$ to form [TiO$_6$]$^{8-}$ octahedron and [FeO$_4$]$^{5-}$ tetrahedra, which increase the degree of depolymerization of the melt.

**Keywords:** fused red mud; molecular dynamics; bridging oxygen; viscosity; SiO$_2$/Al$_2$O$_3$ mass ratio

## 1. Introduction

Fused red mud is the slag formed after iron extraction from red mud (a solid waste formed in the production of alumina) by a rotary hearth furnace. The production of each ton of alumina generates 0.7 to 2.0 tons of red mud [1,2]. It was estimated that the annual red mud emission in 2018 was over 160 tons worldwide with about 105 million tons disposed in China [3]. Fused red mud contains large amounts of inorganic substances and heavy metals, such as CaO, Na$_2$O, As, Cr, Cd, etc. The mass storage of fused red mud is potentially a severe safety hazard and may result in pollution, environmental damage, and a very large waste of resources. Over the past few decades, many methods have been adopted to dispose of red mud. For example, red mud can be used to improve thermal stability and mechanical properties of cement-based grouting materials as well as shorten the setting time and enhance the strength of these materials [4–7]. Additionally, red mud can be added to asphalt mixtures to improve the asphalt performance, bulk density, and rutting resistance [8]. Moreover, red mud can be used as a filler to improve the properties of polyvinyl chloride (PVC) [9] and sisal/polyester composites [10]. Finally, continuous glass fiber can be produced from gold tailings, waste limestone, red mud, and ferronickel slag [11].

Fused red mud is categorized as a silicate and mainly consists of SiO$_2$, Al$_2$O$_3$, MgO, and CaO. The use of red mud to prepare high-quality asbestos fibers is an effective method for recovering red mud. Studies on slag have made great progress. For example, Guo et al. studied the effect of MgO/Al$_2$O$_3$ ratio (from 0.2 to 0.54) and CaO/SiO$_2$ ratio (from 1.05 to 1.35) on the CaO–SiO$_2$–Al$_2$O$_3$–MgO–TiO$_2$ slag system. It was found that increasing the MgO/Al$_2$O$_3$ ratio reduced the liquidus temperature, deformation temperature, flow

temperature, and breakpoint temperature, while the $CaO/SiO_2$ ratio had little effect [12]. Ma et al. added $TiO_2$ to the CMASTF slag system and found that the viscosity first decreased and then increased as the content of $TiO_2$ increased [13]. Liu et al. systematically studied the effects of BaO, MnO, and CaO on the $CaO–SiO_2–MgO–Al_2O_3–BaO–MnO$ slag system. They found that the viscosity of the slag system first increased and then decreased with increasing BaO content; the viscosity decreased with the addition of MnO and CaO [14,15]. As the MgO content in the $MgO–Al_2O_3–TiO_2–CaO–SiO_2$ slag system increased, the viscosity of the slag system decreased, and the fluidity increased [16]. $B_2O_3$ can increase the degree of polymerization of the $CaO–SiO_2–Al_2O_3–MgO$ (CSAM) system, while $Na_2O$, $K_2O$, and MnO have the opposite effect [17]. $CaF_2$ can facilitate the depolymerization of the silicate network and reduce the melting temperature, viscosity, and viscosity activation energy of the glass [18].

Because of the high temperature of molten slag, its atomic motion state is difficult to observe with conventional methods. Studies have shown that molecular dynamics can be useful for studying molten slag, silicate, glass, etc. Zhang et al. [19] used molecular dynamics to study the relationship between the structure and viscosity of the $CaO–SiO_2–Al_2O_3–MgO–TiO_2$ slag system. They found that the degree of $SiO_2$ polymerization is the main factor affecting the viscosity. Mongalo et al. [20] simulated the structure and conductivity of $CaO-MgO-Al_2O_3-SiO_2$ and found that the conductivity can be predicted well under conditions of low basicity. Zhao et al. [21] studied the effect of $CaO/Al_2O_3$ ratio on ladle furnace refining slag. They found that the addition of CaO could introduce charge compensators to promote the polymerization of the slag system, thereby increasing the viscosity of the system. Jiang et al. [22] studied the effects of CaO and MgO on the structure and properties of blast furnace ash and reported that CaO improved the viscosity of blast furnace ash significantly more than MgO. Gao et al. [23] studied the effect of $CaO/Na_2O$ ratio on the diffusivity and melting behavior of slag in the $SiO_2–Al_2O_3–CaO–Na_2O$ system by a new method, ring statistics. Dai et al. [24] studied the effect of CaO on the fusion properties of coal ash and found that CaO could reduce the melting point of coal ash and increase its molten fluidity. Atila et al. [25,26] studied the effects of $Al_2O_3$ on the thermodynamics, elastic properties, and structure of silicate glasses. They found that increasing the $Al_2O_3$ content could increase the population of bridging oxygen (BO) and oxygen triclusters, glass transition temperature, and elastic modulus. Jabraoui et al. [27] found that increasing the $SiO_2$ content in silica calcium aluminosilicate glasses reduced the corresponding elastic constants. Fe ions reduced the degree of polymerization of the $CaO–SiO_2-Fe_tO$ system. $Na_2O–Al_2O_3–SiO_2$ alkali aluminosilicate glass had the best network connectivity when the Al/Na ratio was 1.2 [28,29].

At present, there are few molecular dynamics studies on fused red mud in the molten state. On the other hand, the microstructural properties and viscosity of molten red mud critically influence the quality of the prepared asbestos fibers. Therefore, in this study, the effect of $SiO_2/Al_2O_3$ ratio on the structural properties and viscosity of molten red mud was simulated by molecular dynamics. This work provides theoretical guidance for the preparation of asbestos fibers from fused red mud.

## 2. Simulation Method

The key to the success of the molecular dynamics simulation of fused red mud is to select the appropriate potential form and parameters. The Garofalini potential function was used in this study. The Garofalini potential function is composed of a modified Born–Mayer–Huggins (BMH) potential function and a three-body potential function [30]. The modified BMH potential contains a short-range repulsive term and a modified Coulomb term. The resulting BMH potential function is:

$$V_{ij} = \frac{q_i q_j}{4\pi\varepsilon_0 r_{ij}} + A_{ij}\exp\left(-B_{ij}r_{ij}\right) - \frac{C_{ij}}{r_{ij}^6} - \frac{D_{ij}}{r^8} \tag{1}$$

The first term in the formula represents the Coulomb potential. The molecular dynamics calculation takes into account the interactions not only between any two atoms with residual charges within the cell but also between two atoms with residual charges across cells. The second term in the formula is the short-range repulsive term. The influence of repulsion cannot be ignored in the potential function. Repulsion here mainly refers to the overlap between electron clouds and the electrostatic interaction between nuclei; both can increase the potential energy of the system. The last two terms in the formula represent the attractive potential. The polarization term in the potential function can be neglected. $A_{ij}$ is the charge of atoms $i$ and $j$. $B_{ij}$, $C_{ij}$, and $D_{ij}$ are potential parameters. The potential parameters used in this study are listed in Table 1 [26,31,32].

**Table 1.** Potential parameters in this study.

| $i$ | $j$ | $A_{ij}$ (eV) | $B_{ij}$ (1/Å) | $C_{ij}$ (eV·Å$^6$) |
|---|---|---|---|---|
| O | O | $1.18 \times 10^5$ | 7.31 | 0 |
| Si | Si | $8.32 \times 10^5$ | 8.77 | $2.75 \times 10^1$ |
| Ca | Ca | $2.55 \times 10^5$ | 4.68 | 0 |
| Fe | Fe | $8.49 \times 10^5$ | 6.89 | 0 |
| Mg | Mg | $4.16 \times 10^5$ | 6.75 | $6.91 \times 10^1$ |
| O | Si | $1.98 \times 10^5$ | 5.77 | 0 |
| O | Ca | $1.52 \times 10^5$ | 4.93 | 0 |
| O | Fe | $2.74 \times 10^5$ | 6.08 | 0 |
| O | Mg | $1.06 \times 10^5$ | 4.95 | 0 |
| Si | Ca | $2.43 \times 10^5$ | 5.52 | $7.65 \times 10^0$ |
| Si | Fe | $5.79 \times 10^5$ | 6.64 | $6.56 \times 10^1$ |
| Si | Mg | $5.86 \times 10^5$ | 7.64 | $4.65 \times 10^1$ |
| Ca | Fe | $4.19 \times 10^5$ | 5.68 | 0 |
| Ca | Mg | $1.82 \times 10^5$ | 5.08 | 0 |
| Fe | Mg | $2.52 \times 10^5$ | 5.59 | 0 |
| Al | Al | $2.44 \times 10^3$ | 3.65 | 0 |
| Ti | Ti | $3.52 \times 10^4$ | 6.25 | 0 |
| Na | Na | $1.35 \times 10^3$ | 6.25 | 0 |
| Al | O | $1.95 \times 10^3$ | 3.55 | 0 |
| Ti | O | $2.40 \times 10^5$ | 6.06 | 0 |
| Na | O | $1.99 \times 10^3$ | 6.25 | 0 |
| Na | Si | $1.25 \times 10^3$ | 6.25 | 0 |

In the simulation, the number of atoms of different elements was set according to the mass percentage of various components (Table 2). To ensure that the samples have similar initial conditions, the total numbers of atoms should be as similar as possible. Cubic simulation boxes with a side dimension of 32.5 Å (see Figure 1) were constructed consisting of no less than 2260 and no more than 2280 atoms. The numbers of each atom in the system are listed in Table 3, and the density was set to 2.90 g/cm$^3$.

**Table 2.** Composition of red mud and number of atoms.

| Sample No. | Weight Percentage | | | | | | | $\frac{SiO_2}{Al_2O_3}$ |
|---|---|---|---|---|---|---|---|---|
| | SiO$_2$ | Al$_2$O$_3$ | CaO | MgO | Na$_2$O | Fe$_2$O$_3$ | TiO$_2$ | |
| 1# | 25 | 35 | 12 | 10 | 6 | 8 | 4 | 0.71 |
| 2# | 30 | 30 | 12 | 10 | 6 | 8 | 4 | 1 |
| 3# | 35 | 25 | 12 | 10 | 6 | 8 | 4 | 1.4 |
| 4# | 40 | 20 | 12 | 10 | 6 | 8 | 4 | 2 |
| 5# | 45 | 15 | 12 | 10 | 6 | 8 | 4 | 3 |
| 6# | 50 | 10 | 12 | 10 | 6 | 8 | 4 | 5 |
| 7# | 55 | 5 | 12 | 10 | 6 | 8 | 4 | 11 |

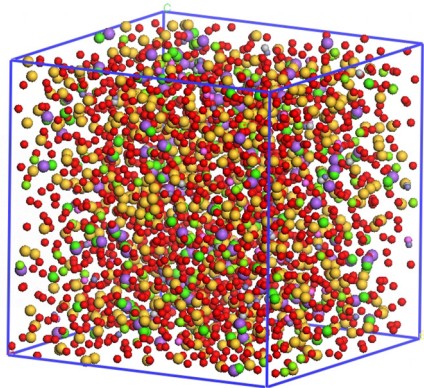

**Figure 1.** Cubic box of a molten red mud system.

**Table 3.** Number of atoms in molecular dynamics.

| Sample No. | Number of Atoms | | | | | | | | |
|---|---|---|---|---|---|---|---|---|---|
| | Ca | Si | Al | Ti | Mg | Fe | Na | O | Total |
| 1# | 107 | 208 | 344 | 25 | 125 | 38 | 96 | 1319 | 2262 |
| 2# | 107 | 250 | 294 | 25 | 125 | 38 | 96 | 1328 | 2263 |
| 3# | 107 | 292 | 246 | 25 | 125 | 38 | 96 | 1340 | 2269 |
| 4# | 107 | 338 | 196 | 25 | 125 | 38 | 96 | 1347 | 2272 |
| 5# | 107 | 375 | 148 | 25 | 125 | 38 | 96 | 1359 | 2273 |
| 6# | 107 | 417 | 98 | 25 | 125 | 38 | 96 | 1368 | 2274 |
| 7# | 107 | 458 | 50 | 25 | 125 | 38 | 96 | 1378 | 2277 |

All simulations were performed in an NVT ensemble with a Nose–Hoover thermostat to ensure that the number of atoms, the volume, and the temperature remained constant throughout the simulation. The initial temperature was set to 5000 K, followed by 30 ps of relaxation to obtain a system with uniform particles. Then, the system was cooled to 2000 K in 30 ps at a cooling rate of $1 \times 10^{14}$ K/s, followed by 30 ps of relaxation at 2000 K. Finally, the system was further cooled to 1800 K in 20 ps at a cooling rate of $1 \times 10^{13}$ K/s, followed by 30 ps of relaxation at 1800 K (see Figure 2). This ensured that the atoms in the system were evenly mixed and the system was kept in the molten state.

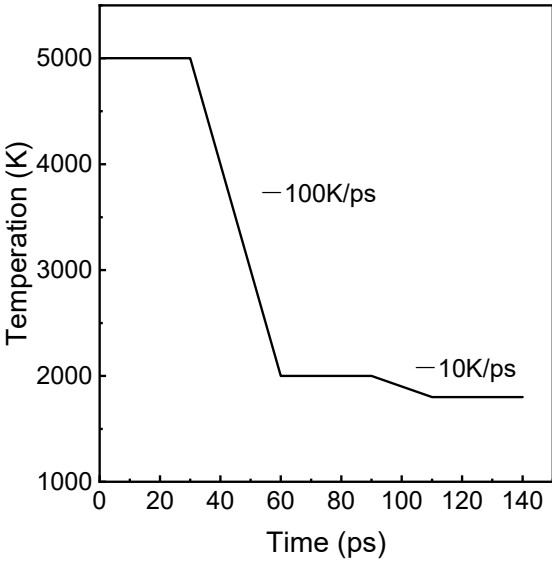

**Figure 2.** Computing process of simulation.

## 3. Results and Discussion

### 3.1. Partial Radial Distribution Function (PDF) and Coordination Number (CN)

The *PDF*, $g_{ij}(r)$ is generally used to assess the short-range order in the metallurgical slags at high temperature which belongs to an amorphous system. The *PDF* can be calculated by Equation (2), where $N_i$ and $N_j$ are the total numbers of ions $i$ and $j$, respectively, $V$ is the volume of the system, and $n(r)$ denotes the average number of the ions $j$ surrounding the ion $i$ in a spherical shell within $r \pm \Delta r/2$. The average coordination number (*CN*), given by Equation (3), can be evaluated by integrating the $g_{ij}(r)$ curve to its first valley.

$$g_{ij}(r) = \frac{V}{N_i N_j} \sum_j \frac{\langle n_{ij}(r - \Delta r/2, r + \Delta r/2) \rangle}{4\pi r^2 \Delta r} \tag{2}$$

$$N_{ij} = 4\pi \frac{N_j}{V} \int_0^r g_{ij}(r) r^2 dr \tag{3}$$

Figure 3a shows the *PDF* curves of Si–O, Al–O, Ca–O, Mg–O, Na–O, Ti–O, and Fe–O of the fused red mud sample #1. The first peak of the curve indicates the average bond length between the two atoms. The bond lengths for different pairs are 1.59 Å, 1.79 Å, 2.31 Å, 2.29 Å, 2.33 Å, 2.45 Å, and 2.51 Å, respectively. The calculated results are consistent with those previously determined by molecular dynamics simulations and experiments [33–36]. The bond lengths of Si–O and Al–O are similar and much smaller than those of Ca–O and Mg–O. Compared with other T–O (T = Al, Ca, Mg, Na, Ti, and Fe) bonds, the Si–O bond shows the strongest first peak, indicating that the Si–O bond is the most stable.

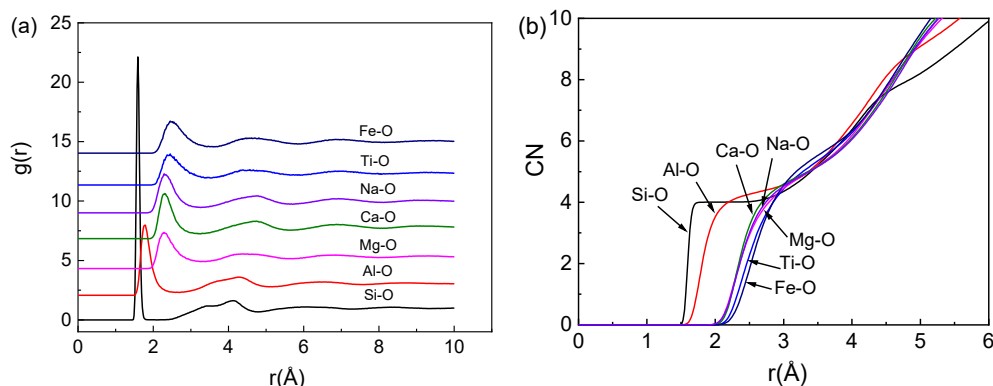

**Figure 3.** (**a**) PDF and (**b**) CN of sample #1 in the fused red mud system.

Figure 3b shows the CNs of different atom pairs in the sample. The value corresponding to the CN platform is the average coordination number. The CN of Si–O remains basically unchanged at approximately 4. The CN platform of Al–O is not as flat as that of Si–O, suggesting that the Al–O bond is slightly less stable than the Si–O bond. Except for Si–O and Al–O, the other T–O bonds do not form platforms, indicating that these T–O bonds do not form a stable network structure.

### 3.2. Distribution of Bond Angles

There are two types of bond angles in the fused red mud. One is the O–T–O (T = Si, Al) bond angle in the tetrahedron, and the other is the T–O–T bond angle by which the tetrahedra are connected to each other in the network. The shape characteristic of the tetrahedral structure can be determined by the distribution of O–T–O bond angles. The T–O–T bond angle represents the direction in which the tetrahedra are connected in the glass network structure. The flexibility of the T–O–T bond angle determines the degree of disorder of the melt, and it varies in a wide range (approximately 120~180°). The change in the distribution of the T–O–T bond angle indicates whether the structure has become more ordered or disordered.

Figure 4a shows that with increasing $SiO_2/Al_2O_3$ ratio, the peak value of the O–Si–O bond angle distribution increases, and the peak width decreases. The bond angle is approximately 110.2°, and the theoretical bond angle of tetrahedral O–Si–O is 109.5° [36,37]. The result suggests that the structure of $[SiO_4]^{4-}$ becomes more stable with increasing $SiO_2/Al_2O_3$ ratio. Figure 4b shows the distribution of the O–Al–O bond angle in the aluminum oxide tetrahedron. The bond angle is approximately 106.5°, and the peak is wider than that of O–Si–O, indicating that $[AlO_4]^{5-}$ has a more regular shape but slightly lower stability than $[SiO_4]^{4-}$. Figure 4c shows the Si–O–Si bond angle distribution. The bond angle is approximately 144.5°. The width of the peak gradually decreases as the $SiO_2/Al_2O_3$ ratio increases, indicating that more $[SiO_4]^{4-}$ tetrahedra are connected with each other and that the degree of order increases. Figure 4d shows the distribution of the Al–O–Si bond angle. The bond angle is approximately 138.9°. As the $SiO_2/Al_2O_3$ ratio increases, the peak intensity gradually decreases, but the peak width becomes larger, indicating that the connection between $[SiO_4]^{4-}$ tetrahedron and $[AlO_4]^{5-}$ tetrahedron decreases and that the structure becomes less ordered [38].

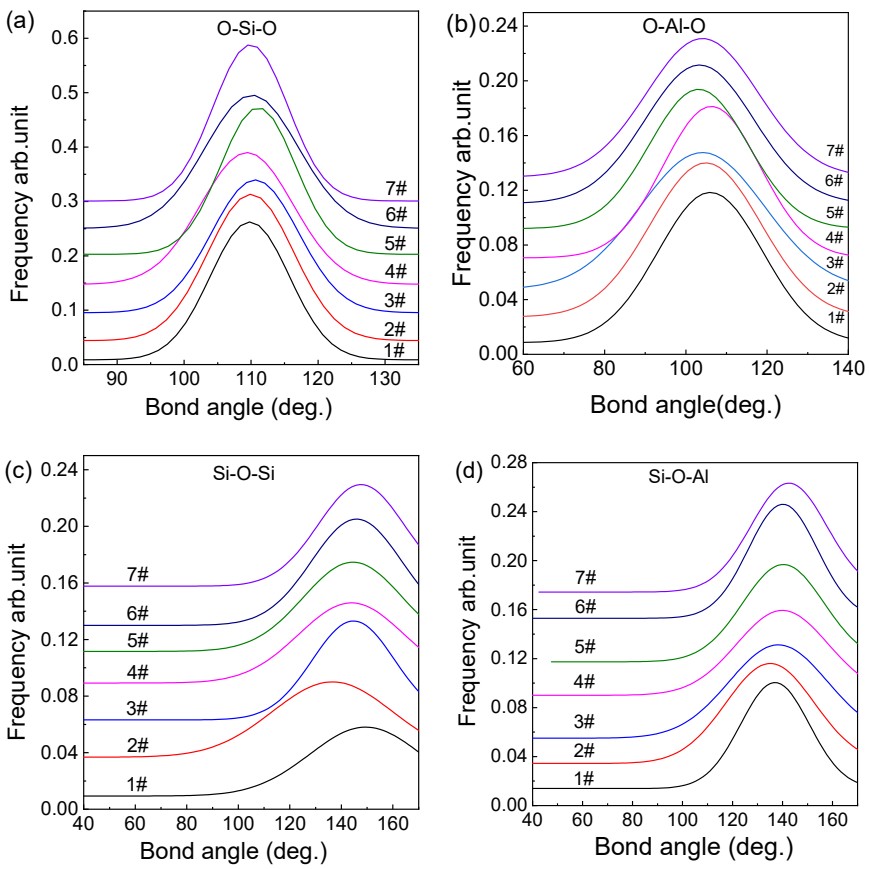

**Figure 4.** Distribution of bond angles of (**a**) O–Si–O, (**b**) O–Al–O, (**c**) Al–O–Al, and (**d**) Al–O–Si.

### 3.3. Distribution of Oxygen and $Q^n$

The tetrahedral structures of the network are not connected by edges or faces, only by corners. The oxygen at the shared corner is the BO, which connects two tetrahedral structures. The oxygen in a tetrahedral structure but not connected to other tetrahedral structures is a nonbridging oxygen (NBO) [21,26,37]. The network structure of fused red mud consists of different tetrahedral structures. $Si^{4+}$, $Al^{3+}$, $Ti^{4+}$, and $Fe^{3+}$ ions can form $[SiO_4]^{4-}$ and $[AlO_4]^{5-}$ tetrahedral, $[TiO_6]^{8-}$ octahedron, and $[FeO_4]^{5-}$ tetrahedral structures with oxygen [36], respectively. Si and Al are network formers, and their ionic groups $[SiO_4]^{4-}$ and $[AlO_4]^{5-}$ are the main components of the network.

Figure 5 shows the distribution of BO and NBO of molten red mud with different $SiO_2/Al_2O_3$ mass ratios. Figure 5a shows that when the ratio of $SiO_2/Al_2O_3$ is small, the concentration of BO follows the order Si–O–Al>Al–O–Al>Si–O–Si. As the ratio of $SiO_2/Al_2O_3$ increases, the concentration of Si–O–Si BO increases correspondingly, and the concentration of Al–O–Al BO decreases rapidly, whereas the concentration of Si–O–Al BO first increases and then decreases. Figure 5b shows that with increasing $SiO_2/Al_2O_3$ ratio, the concentrations of Si–O–Ca NBO, Si–O–Mg NBO, and Al–O–Na NBO increase slightly, and the concentrations of the rest NBOs are almost unchanged.

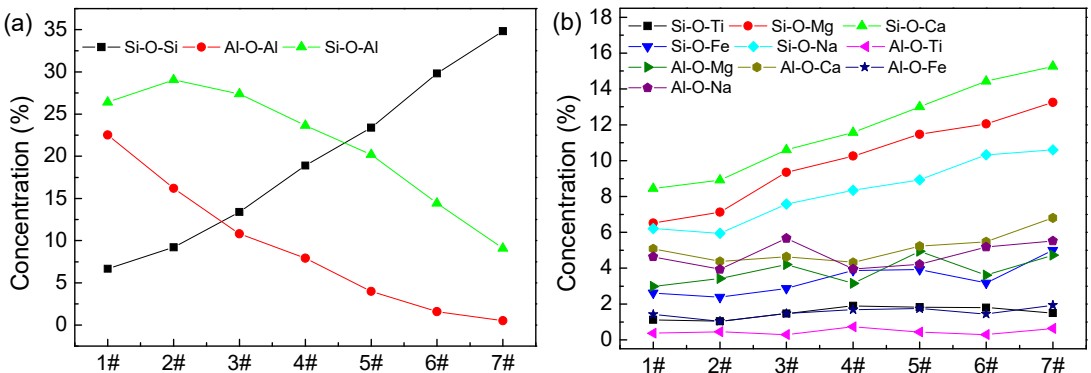

**Figure 5.** Distribution of BO and NBO in molten red mud: (**a**) BO and (**b**) NBO.

This phenomenon conforms to the Al avoidance principle [38]; i.e., Si–O–Al has the lowest bond energy, followed by Si–O–Si, and Al–O–Al has the highest bond energy. When the ratio of $SiO_2/Al_2O_3$ is small, the content of $Si^{2+}$ in the molten red mud is low, and the content of $Al^{3+}$ is high. As the $Si^{2+}$ content increases, more Si–O–Al bonds are formed in the melt according to the Al avoidance principle. With increasing $Si^{2+}$ and decreasing $Al^{3+}$, there is insufficient $Al^{3+}$ in the melt to form Si–O–Al bonds. As a result, the content of Si–O–Al begins to decrease, thereby accelerating the formation of the Si–O–Si bonds.

Figure 6 shows the distribution of $Q_m^n$ in [SiO$_4$] and [AlO$_4$] tetrahedra, where m (m = Si, Al) represents the atoms connected to BO in the $[SiO_4]^{4-}/[AlO_4]^{5-}$ tetrahedra and $n$ represents the number of Bos in each $[SiO_4]^{4-}/[AlO_4]^{5-}$ tetrahedron. Figure 6a shows that, as the $SiO_2/Al_2O_3$ ratio increases, the concentrations of $Q_{Si}^0$ and $Q_{Si}^1$ in the $[SiO_4]^{4-}$ tetrahedra gradually decrease, the concentration of $Q_{Si}^2$ first increases and then decreases, and the concentrations of $Q_{Si}^3$ and $Q_{Si}^4$ gradually increase. In addition, $Q_{Si}^4$ is produced only in a small amount when the ratio of $SiO_2/Al_2O_3$ is 1.4.

Figure 6b shows that as the $SiO_2/Al_2O_3$ ratio increases, the $Q_{Al}^0$ concentration in the [SiO$_4$] tetrahedra first changes slowly and then increases rapidly; the concentration of $Q_{Al}^1$ first increases and then decreases; and the concentrations of $Q_{Al}^2$ and $Q_{Al}^3$ first change slowly and then decrease rapidly. The amount of $Q_{Al}^4$ is very small. Moreover, $Q_{Al}^4$ disappears when the ratio of $SiO_2/Al_2O_3$ is greater than 3.

As shown in Figure 6c, with increasing $SiO_2/Al_2O_3$ ratio, the $Q_{Al}^0$ concentration in the $[AlO_4]^{5-}$ tetrahedra increases rapidly from the initial 17.44% to 76%. However, the concentrations of $Q_{Al}^1$ and $Q_{Al}^2$ increase first and then decrease; the amount of $Q_{Al}^4$ remains very small and disappears when the $SiO_2/Al_2O_3$ ratio is greater than 3.

As shown in Figure 6d, $Q_{Si}^0$ and $Q_{Si}^1$ are the main species in the $[AlO_4]^{5-}$ tetrahedra. With increasing $SiO_2/Al_2O_3$ ratio, the concentration of $Q_{Si}^0$ decreases from 83.1% to 68%, while the concentration of $Q_{Si}^1$ increases from 15.1% to 30%. The amounts of $Q_{Si}^2$, $Q_{Si}^3$, and $Q_{Si}^4$ remain very small and are minimally affected by the ratio of $SiO_2/Al_2O_3$.

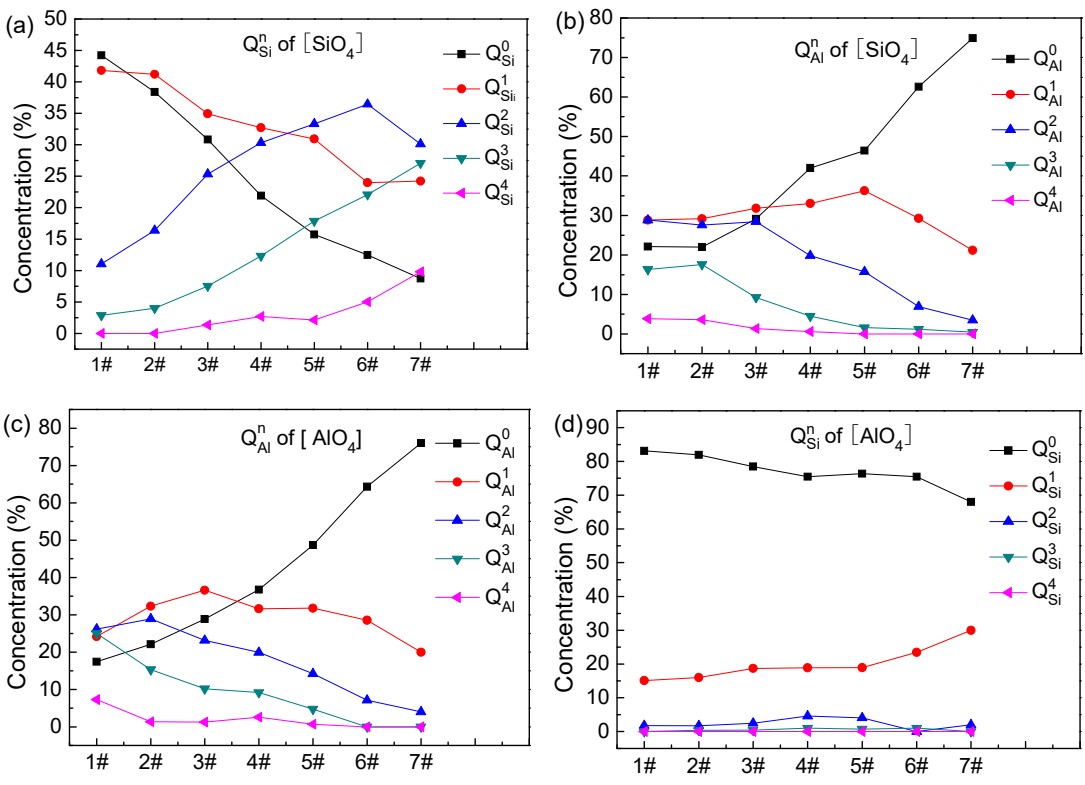

**Figure 6.** Distribution of $Q^n$ species: (**a**,**c**) [SiO$_4$] tetrahedron, (**b**,**d**) [AlO$_4$] tetrahedron.

### 3.4. Self-Diffusion Coefficient and Viscosity

The atoms in the molten red mud are always moving, and the movement of each atom directly affects the viscosity of the melt. Viscosity is a critical parameter for the preparation of fibers using fused red mud and can directly affect the quality of the fibers. The mean square displacement (MSD) of the particles was calculated by analyzing the trajectories of the particles using the molecular dynamics method, and the self-diffusion coefficient of each atom was obtained by combining with the Einstein diffusion equation [39], as shown in Equations (4) and (5):

$$MSD = \langle \Delta r(t)^2 \rangle = \frac{1}{N} \langle \sum_{i=1}^{N} \left| r_{i(t)} - r_{i(0)} \right|^2 \rangle \tag{4}$$

where $r_{i(0)}$ is the position vector of atom $i$ at time 0; $r_{i(t)}$ is the position vector of atom $i$ at time $t$;

$$D = \lim_{t \to \infty} \frac{1}{6} \frac{d\left[ \Delta \bar{r}(t)^2 \right]}{dt} = \lim_{t \to \infty} \frac{1}{6} \frac{d[MSD]}{dt} \tag{5}$$

Figure 7 shows the typical MSD curves as a function of time. Figure 6 shows the effect of SiO$_2$/Al$_2$O$_3$ mass ratios on the diffusion coefficient of each atom in the fused red mud system. Figure 8 shows that the ratio of SiO$_2$/Al$_2$O$_3$ has different effects on the diffusion of different atoms. The $D_{Mg}$ value of Mg is much larger than those of other atoms and decreases with increasing SiO$_2$/Al$_2$O$_3$ ratio. As the SiO$_2$/Al$_2$O$_3$ ratio increases, $D_{Ti}$ first decreases, then increases, and finally decreases again; whereas $D_{Fe}$ remains unchanged at first, then rapidly decreases, and finally stabilizes. The diffusion coefficients of the remaining atoms decrease slowly with increasing SiO$_2$/Al$_2$O$_3$ ratio and follow the order $D_{Na} > D_{Ca} > D_{Al} > D_O > D_{Si}$. Mg is mainly present in the gaps of the network structure formed by silicon and aluminum. It is a network modifier and does not participate in the formation of the network. Thus, Mg can move in a large area, facilitating its diffusion. As for Ti, part of the Ti$^{4+}$ can combine with O$^{2-}$ to form [TiO$_6$]$^{8-}$ octahedron and participate in the formation of network structure. When the content of Al$_2$O$_3$ in the system is relatively

high, $Ti^{4+}$ is mainly present in the gaps of the network structure and has high diffusivity. When the content of $Al_2O_3$ decreases and the content of $SiO_2$ is low, part of $Ti^{4+}$ combines with $O^{2-}$ to form $[TiO_6]^{8-}$ octahedron and participates in network formation, resulting in decreased $D_{Ti}$ value. However, as the amount of $SiO_2$ increases, the binding between $Si^{4+}$ and $O^{2-}$ is stronger than that between $Ti^{4+}$ and $O^{2-}$, so $Ti^{4+}$ is squeezed out of the network structure, leading to an increase in its MSD. With the further addition of $SiO_2$ and decrease in $Al_2O_3$, the viscosity of the system increases, resulting in difficulty in the migration of $Ti^{4+}$ and thereby a decrease in MSD. Similarly, the $Fe^{3+}$ in the gaps of the network structure also has strong diffusivity. Moreover, some $Fe^{3+}$ participates in the formation of the silicate network to form $[FeO_4]$, reducing the movement range of $Fe^{3+}$. As the ratio of $SiO_2/Al_2O_3$ increases, the content of $Si^{2+}$ in the system increases. The binding between $Si^{4+}$ and $O^{2-}$ is stronger than that between $Fe^{3+}$ and $O^{2-}$, so $Fe^{3+}$ is squeezed out of the network structure, resulting in an increase in its $D$ value. However, as the ratio of $SiO_2/Al_2O_3$ further increases, more $[SiO_4]$ tetrahedra are formed, and the network becomes more complete. The viscosity of the system increases accordingly, the migration of $Fe^{3+}$ is restricted, and its $D$ value decreases. $Si^{4+}$ and $Al^{3+}$ combine with $O^{2-}$ to form $[SiO_4]^{4-}$ and $[AlO_4]^{5-}$ tetrahedral structures. They are network formers and do not easily diffuse.

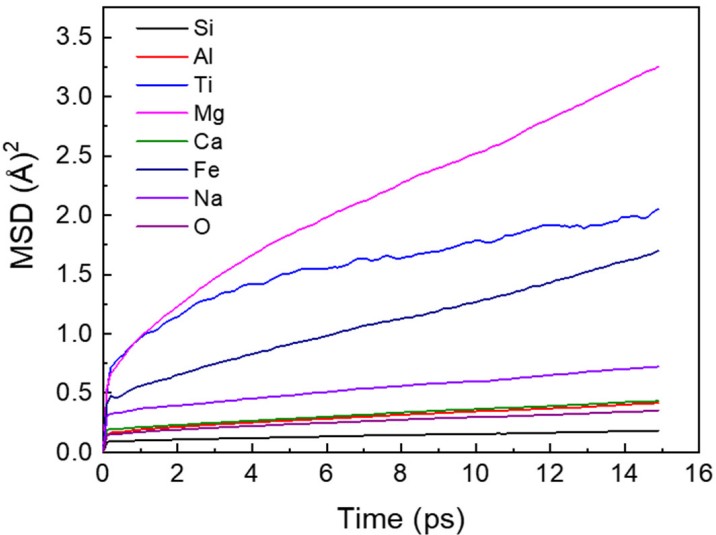

**Figure 7.** MSD curves for the #1 fused red mud system.

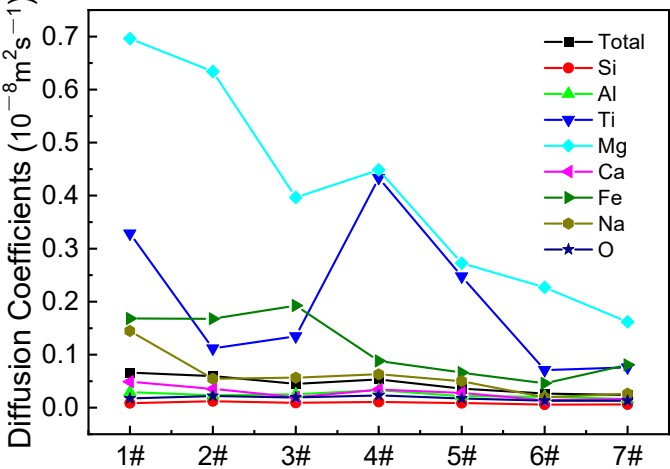

**Figure 8.** Effect of $SiO_2/Al_2O_3$ ratio on diffusion coefficients.

Equations relating viscosity to diffusivity have been widely used in the study of silicate melts [26,31,40], and the two most popular equations are the Eyring equation and the Stokes–Einstein equation. They were derived in different ways but relate diffusivity and viscosity through the same formula, as shown in Equation (6).

$$\eta = \frac{K_B T}{D\lambda} \qquad (6)$$

where $\eta$ is the viscosity coefficient, J·m$^{-1}$·s; $K_B$ is the Boltzmann constant; T is the $T_{end}$, K; and $\lambda = 2R$, where R is the average diffusion distance of ions, Å. Based on this, the viscosity of the molten red mud is calculated by the total diffusion coefficient of the ions in the system. Figure 9 shows the effect of $SiO_2/Al_2O_3$ ratio on the viscosity and the total diffusion coefficient of molten red mud, the orange red histogram and blue curve in the figure represent the change of viscosity $\eta$ and total diffusion coefficient $D_{Total}$, respectively, as can be seen from the figure, with the increasing $SiO_2$, the viscosity first increase, then decrease slightly, and finally increase again. Conversely, the change of total diffusion coefficient is opposite. As discussed above, the bonding between $Si^{4+}$ and $O^{2-}$ is stronger than that between $Ti^{4+}$ and $O^{2-}$ or $Fe^{3+}$ and $O^{2+}$. Whin the content of $SiO_2$ is low, part of $Ti^{4+}$and $Fe^{3+}$ combines with $O^{2-}$ to form $[TiO_6]$ octahedron and $[FeO_4]$ tetrahedra, respectively, and participate in network formation resulting in decreased the values of $D_{Total}$. However, with the gradual increase of $Si^{4+}$, $Ti^{4+}$ and $Fe^{3+}$ are squeezed out of the network structure, at this time, $Ti^{4+}$ and $Fe^{3+}$ have strong diffusivity, leading to an increase in its $D_{Total}$ and decrease in its viscosity, as show in Figure 9. When the $SiO_2/Al_2O_3$ ratio increases from 1.4 (3# sample) to 2 (4# sample), $D_{Total}$ value increase from $4.5 \times 10^{-10}$ m$^2$s$^{-1}$ to $5.3 \times 10^{-10}$ m$^2$s$^{-1}$, correspondingly, the viscosity $\eta$ value drops from 1.27 Pa·s to 1.06 Pa·s. However, as the amount of $Si^{2+}$ in the melt further increases, more $[SiO_4]$ tetrahedra are formed, the degree of polymerization of the network is higher, the diffusion of $Ti^{4+}$ and $Fe^{3+}$ is limited again, and the $D_{Total}$ becomes smaller, from $5.3 \times 10^{-10}$ m$^2$s$^{-1}$ (4# sample) to $2.4 \times 10^{-10}$ m$^2$s$^{-1}$ (7# sample), correspondingly, the $\eta$ increases again, from 1.06 Pa·s to 2.37 Pa·s.

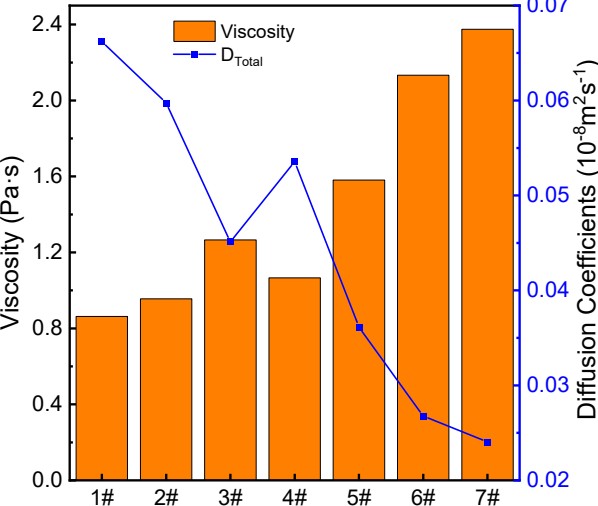

**Figure 9.** Effect of $SiO_2/Al_2O_3$ ratio on viscosity and total diffusion coefficients.

## 4. Conclusions

The effect of $SiO_2/Al_2O_3$ mass ratio on the structural properties and viscosity of the molten fused red mud were systematically analyzed by the molecular dynamics method. The conclusions are as follows:

1.  The average bond lengths of Si–O, Al–O, Ca–O, Mg–O, Na–O, Ti–O, and Fe–O are 1.59 Å, 1.79 Å, 2.31 Å, 2.29 Å, 2.33 Å, 2.45 Å, and 2.51 Å, respectively. The network formers of the fused red mud are $[SiO_4]$ and $[AlO_4]$ tetrahedra.

2.  With increasing $SiO_2/Al_2O_3$ ratio, the concentration of Si–O–Si BO in the molten red mud increases, the concentration of Al–O–Al BO decreases, and the concentration of Si–O–Al BO first increases and then decreases; the concentrations of Si–O–Ca NBO, Si–O–Mg NBO, and Si–O–Na NBO increase, and the remaining NBOs do not change significantly.

3.  The average bond angles of O–Si–O, O–Al–O, Si–O–Si, and Al–O–Si in the fused red mud are $110.2°$, $106.5°$, $144.5°$, and $138.9°$, respectively. With increasing $SiO_2/Al_2O_3$ ratio, the peaks of O–Si–O and Si–O–Si bond angle distributions become narrower, while the peaks of O–Al–O and Al–O–Si become wider. The results indicate that the degree of polymerization of $[SiO_4]$ tetrahedral network increases and that the degree of polymerization of $[AlO_4]$ tetrahedra decreases, leading to a more compact melt structure.

4.  With increasing $SiO_2/Al_2O_3$ ratio, (1) in the distribution of $Q_{Si}^n$ of $[SiO_4]$, $Q_{Si}^0$ and $Q_{Si}^1$ decrease, and $Q_{Si}^2$ first increases and then decreases; (2) in the distribution of $Q_{Al}^n$ of $[SiO_4]$, $Q_{Al}^0$ increases, $Q_{Al}^1$ first increases and then decreases, $Q_{Al}^2$ and $Q_{Al}^3$ first change slowly and then decrease, and the content of $Q_{Al}^4$ is very small and goes to zero when the ratio of $SiO_2/Al_2O_3$ is greater than 3; (3) the concentration of $Q_{Al}^0$ in the $[AlO_4]$ tetrahedra increases rapidly from the initial 17.44% to 76%, $Q_{Al}^1$ and $Q_{Al}^2$ first increase and then decrease, the content of $Q_{Al}^4$ remains very small and goes to zero when the ratio of $SiO_2/Al_2O_3$ is greater than 3; and (4) the concentration of $Q_{Si}^0$ in the $[AlO_4]$ tetrahedra decreases from 83.1% to 68%, while the concentration of $Q_{Si}^1$ increases from 15.1% to 30%, and the concentrations of $Q_{Si}^2$, $Q_{Si}^3$, and $Q_{Si}^4$ remain very small and are minimally influenced by the $SiO_2/Al_2O_3$ ratio.

5.  With increasing $SiO_2/Al_2O_3$ ratio, the viscosity of the fused red mud first increases, then decreases, and finally increases because $Ti^4$ and $Fe^{3+}$ combine with $O^{2-}$ to form $[TiO_4]$ octahedron and $[FeO_4]$ tetrahedra, which increase the degree of polymerization of the melt.

**Author Contributions:** Y.L. and B.X. performed theoretical analysis and wrote the article; Y.C. and Z.W. performed the simulation calculations; and P.D. helped with data collation. All authors have read and agreed to the published version of the manuscript.

**Funding:** This study was funded by the China National Natural Science Foundation Regional Innovation and Development Joint Fund (Grant No. U20A20271) and the China National Natural Science Foundation General Program (Project No. 51874138).

**Data Availability Statement:** Data of this study are available on request from the corresponding authors.

**Conflicts of Interest:** The authors declare no conflict of interest.

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
