# Peer review of "Molecular Dynamics Study on the Effect of SiO2/Al2O3 Mass Ratio on the Structural Properties and Viscosity of Molten Fused Red Mud"

_minerals, doi:10.3390/min12080925_

Round 1

Reviewer 1 Report

The manuscript entitled "Molecular dynamics study on the effect of SiO2/Al2O3 mass ratio on the structural properties and viscosity of molten fused red mud" is an interesting work on molten fused red mud for effective utilization of waste and circular economy. The authors propose an in-depth study of molecular dynamics using Garofalini potential function. The Pair distribution function (PDF) and coordination number (CN) reported may be supported by studies reported earlier. The distribution of bond angle studies needs to be explained based on earlier studies reported by other researchers.

Author Response

dear expert

Reviewer 2 Report

This paper reports the effect of SiO2/Al2O3 mass ratio on the structural properties and viscosity of molten fused red mad based on the molecular dynamics study. The authors highlight the correlation between the degree of polymerization of the network and the viscosity of the melt. I agree well with significance of the paper, but there are several points to be considered prior to publication.

 1. I could not understand well the calculation conditions. How about the cell size? Besides, please comment on time step width and total time. Movie data is available?

 2. I am interested in the coordination number (CN) of Ti and Fe in the network. Especially for the sample #7. The authors should compare and discuss the coordination mode of these Ti and Fe between #1 and #6, #7. The CN of Ti and Fe are both four? In general, the CN of Ti and Fe is six.

Author Response

Dear expert

Reviewer 3 Report

Dear Authors,

Your paper “Molecular dynamics study on the effect of SiO2/Al2O3 mass ratio on the structural properties and viscosity of molten fused red mud” models some specific parameters such as number of bridging and non-bridging oxygens, coordination number and viscosity of fused red muds by varying the SiO2/Al2O3 ratio.

The topic is of a fundamental importance to correct design reduction and smelting process of red mud reduction for iron recovery since one of the main issues is the correct separation between iron and slag fraction due to the excess of slag viscosity. Since the paper takes into account the main chemical species within the slag fraction (fused red mud) it has a high and reliable applicability.

The paper is well organized and the modelling procedure is described with enough details. Results are presented in a comprehensible way and the discussion suffers of a lack of comparison with existing literature on the same topic.

Although, the paper has all the requirements to be published in Minerals journal, it needs some minor revision to reach the desired level of quality.

Please find in attachment a list of comments/remarks/corrections for improving your work.

P.1, line 27-28: “…and the annual output and total output of red mud in China are very large.” Please instead to be so generic, specify the amount of annual and total red mud production in China and, eventually, all around the world.

P.1, line 28-29: “Fused red mud contains large amounts of 28 inorganic substances and heavy metals.”. It would be better to specify which toxic metals they are, especially because if the red muds were reduced to separate iron, the most of toxic elements (Cr, Ni, As, V, Mo, W, …) went to the iron phase and only few of them (Ba for instance) remained in the fused RM.

P.2 line 58-59: “Because of the high temperature of molten slag, its microstructure is difficult to observe with conventional methods.”. In which sense? What the Authors would mean? Please clarify this point

P.3, Table 1: Why the couple “Al-Al”, “Ti-Ti”, “Al-O” and “Ti-O” and their interactions were not reported? I supposed that those parameters are really important since Al2O3 and TiO2 concentration in fused red mud are not negligible. The same for Na.

P.4, Table 3: what is MD in the caption? Please specify.

P.4, Figure 1: Please specify on X- and Y-axes what the parameters are. In other words, what is r(Å)? I suppose is the radius of the pair. What is g(r)? Can the Authors also explain how and why for a fixed pair, r(Å) can vary?

P.4, line 127: “Figure 1 shows the pair distribution function (PDF)…”. I suppose Figure 1 is Figure 1a

P.4, line 136: “Figure 2 shows the coordination numbers (CNs)…”. I suppose Figure 2 is Figure 1b

P.5, Figure 2: in figure 2b the curve related to mix #7 is not reported. Why? It is an error it depends on the chemical composition of the mix?

P.8, equation 3: Please specify the parameters in equation 3

P.9, line 267-282. The part related to viscosity is the less discussed if compared with the other aspects of the fuse red mud. I warmly suggest to improve the discussion on viscosity variation in the same way done for other parameters.

P.10, conclusion n. 5: “With increasing SiO2⁄Al2O3 ratio, the viscosity of the fused red mud first increases, then decreases, and finally increases because Ti4 and Fe3+ combine with O2- to form [TiO4] tetrahedra and [FeO4] tetrahedra, which reduce the degree of polymerization of the melt.” I think the last part of the sentence is “which reduce the degree of depolymerization of the melt”, since the viscosity at high SiO2/Al2O3 ratio increases and the polymeric network re-incorporate the [TiO4] and [FeO4] tetrahedra. Please check and correct accordingly.

Best Regards

Author Response

Dear expert
